

# The Super-Turbine Wind Power Conversion Paradox: Using Machine Learning to Reduce Errors Caused by Jensen's Inequality

Tyler C McCandless[1], Sue Ellen Haupt[1]

[1]National Center for Atmospheric Research, 3450 Mitchell Lane, Boulder, CO 80301 USA

5   *Correspondence to*: Tyler C. McCandless (mccandle@ucar.edu)

**Abstract.** Wind power is a variable generation resource and therefore requires accurate forecasts to enable integration into the electric grid. Generally, the wind speed is forecast for a wind plant and the forecasted wind speed is converted to power to provide an estimate of the expected generating capacity of the plant. The average wind speed forecast for the plant is a function of the underlying meteorological phenomena being predicted; however, the wind speed for each turbine at the farm 10 is also a function of the local terrain and the array orientation. Conversion algorithms that assume an average wind speed for the plant, i.e., the super-turbine power conversion, assume that the effects of the local terrain and array orientation are insignificant in producing variability in the wind speeds across the turbines at the farm. Here, we quantify the differences in converting wind speed to power at the turbine level compared to a super-turbine power conversion for a hypothetical wind farm of 100 2-MW turbines as well as from empirical data. The simulations with simulated turbines show a maximum 15 difference of approximately 3% at 11 m s$^{-1}$ with 1 m s$^{-1}$ standard deviation of wind speeds and 8% at 11 m s$^{-1}$ with 2 m s$^{-1}$ standard deviation of wind speeds as a consequence of Jensen's Inequality. The empirical analysis shows similar results with mean differences between converted wind speed to power and measured power of approximately 68 kW per 2 MW turbine. However, using a random forest machine learning method to convert to power reduces the error in the wind speed to power conversion when given the predictors that quantify the differences due to Jensen's Inequality. These significant 20 differences can lead to wind power forecasters over-estimating the wind generation when utilizing a super-turbine power conversion for high wind speeds, and indicates that power conversion is more accurately done at the turbine level if no other compensatory mechanism is used to account for Jensen's Inequality.

## 1 Introduction

As the capacity of renewable energy resources increases, accurate forecasts of power production are becoming increasingly 25 instrumental for efficient and effective management of the energy grid. In 2017, the worldwide wind power capacity grew by 10.8% to a total capacity of 539 GW (World Wind Energy Association 2018). This capacity covers only about 5% of the total global energy demand, so continued growth of wind power generation capacity is expected. Large wind power plants that have tens to hundreds of turbines pose many challenges for forecasting as meteorological conditions, the topography, array orientation, and resulting wake effects may affect wind and power variability across the turbines at the farm. 30 Ultimately, the variability in the wind power needs to be accounted for in farm-level, day-ahead wind power forecasts that are used in unit commitment and electricity market bidding strategies, as well as for intra-day wind power forecasts that are used for reliability, regulation, or sales on the spot market (Ahlstrom et al. 2013, Orwig et al. 2014).

There are two main sources of error in wind power forecasting: the error in the underlying weather forecast (i.e., wind speed, 35 and to a lesser degree air density) and the error in converting the wind speed to power. Past research has indicated an advantage in using artificial intelligence methods for wind power conversion (Parks et al. 2011), and we further investigate this in the context of the super-turbine approach. In the super-turbine conversion methodology, the wind speed is forecast as a farm-average value, and that wind speed is converted to farm-level power. Bartlett (2018) pointed out that the super-turbine approach can result in substantial errors in power conversion, especially when variability exists across a wind farm. 40 He further analyzed wind farm data to explore alternate methods to convert wind speed to power. The underlying assumption



of the super-turbine approach ignores the variability in wind speed across the turbines and the nonlinearity of the power curve. Some methods, however, have added wind variability as an explanatory variable in wind power conversion (Ritter et al. 2015). The issue of wind farm variability is illustrated in Figure 1, where the blue line indicates the power conversion for the super-turbine (i.e., farm-level mean wind speed), while the red distribution illustrates that each turbine may have

different wind speeds centered around the mean value, which results in a distribution of power with a lower mean value than the super-turbine for a 10 m s$^{-1}$ wind speed. For this notional example, the super-turbine approach would predict an average power of approximately 1600 kW while the turbine-level approach would predict approximately 1500 kW.

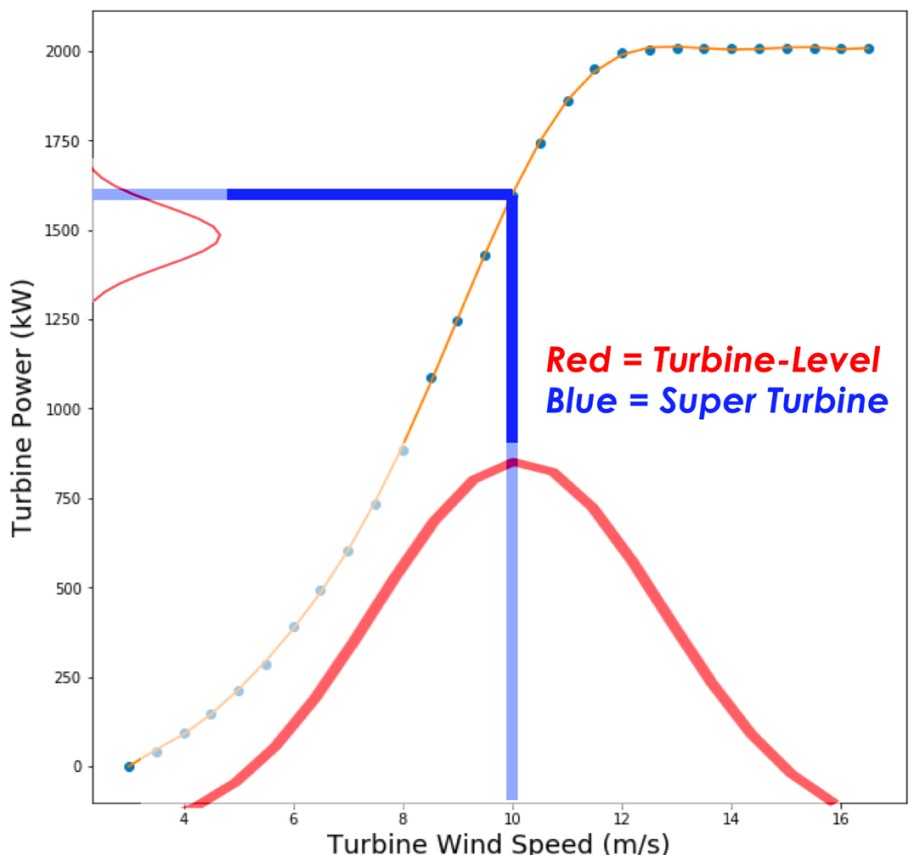

**Figure 1:** Illustration of an instance for converting wind speed to power for a number of turbines and for the average wind speed across the turbines, which is termed the super-turbine power conversion methodology. This shows that the mean power converted from the distribution of wind speeds at the turbines is less than the super-turbine power conversion for a mean wind speed of 10 m s$^{-1}$.

The explanation for this phenomenon is in Jensen's Inequality, which states that the convex transformation of a mean is less than or equal to the mean applied after convex transformation, and vice versa for a concave transformation (Jensen 1906).

The steep portion of the curve for converting wind speed to wind power is generally taken as cubic following the power density function. Thus, at low wind speed values the transformation is convex and at high wind speed values the transformation is concave, which is illustrated in Figure 1 by the orange line. Therefore, at low wind speeds we expect the super-turbine power conversion (i.e., the mean applied before) to be less than the turbine-level power conversion but at high wind speeds we expect the super-turbine power conversion to be greater than the turbine level power conversion. The

application of Jensen's Inequality to wind power conversion is herein described as the super-turbine wind power conversion paradox.



Our goal is to quantify the error due to Jensen's Inequality for a super-turbine power conversion using both simulated hypothetical wind farm data and empirical data. We show the expected difference using simulations for a hypothetical wind farm that has 100 turbines and empirical data from the Shagaya Renewable Energy Park in western Kuwait. In section 2, we describe the methodology for the simulation of the hypothetical wind farm and present results. In section 3, we present and
discuss the impact of Jensen's Inequality in an empirical analysis for the 10-MW Shagaya wind farm. In section 4, we discuss the results of the hypothetical and empirical analysis. In section 5, we propose a machine learning technique for predicting the total wind farm power and discuss the impact of those results in overcoming Jensen's Inequality. Section 6 presents conclusions and suggestions for next steps.

## 2 Hypothetical Wind Farm

### 2.1 Simulation Methodology for a Hypothetical Wind Farm

We statistically simulate wind speeds for a hypothetical wind farm to quantify the expected differences for turbine-level and farm-level power conversions for a variety of theoretical meteorological conditions. Our hypothetical wind farm has 100 2-MW wind turbines, and we simulated these 100 turbine wind speeds 1000 times for each mean wind speed considered. To do this, we sampled from a Gaussian distribution with multiple different mean wind speeds and two different wind speed
standard deviations. We tested mean wind speeds of 6, 7, 8, 9, 10, 11, 12, and 13 m s$^{-1}$ with a standard deviation of 1 m s$^{-1}$, and we tested the same mean wind speeds with a standard deviation of 2 m s$^{-1}$. The standard deviation represents the variability of wind speeds across the wind turbines at the farm that would be affected by the meteorological conditions, the topography and array orientation. Although one would expect a general wind speed distribution to be best fit with a Weibull distribution, here we use a Gaussian because we are sampling from specific points designed as the wind farm's mean wind
speed in the overall distribution of wind speeds. We also opted for a Gaussian distribution because the variability across the turbines at a wind farm are caused by multiple factors including turbulence, wake effects, local terrain, turbine mechanics and other micro-scale weather, which is a different underlying driver of variability than using a Weibull distribution to characterize the long-term climatology of wind speeds at a farm.

Next, we convert the wind speed to power for the turbine-level and farm-level wind speed for each simulation using a typical power curve. For our power conversion methodology, we use a 10th-order polynomial fit to 2 MW Vestas turbine data (Vestas 2018). The 10th-order polynomial fit adequately captures the convex shape below 9 m s$^{-1}$ and the concave nature of the cubic wind speed to power conversion for wind speeds above 9 m s$^{-1}$, as illustrated in Figure 2. Although wind turbine manufacturers typically provide power curves under ideal conditions while turbines operate in a wide variety of
meteorological conditions that are seldom ideal, this conversion is standard and serves our purpose of having a consistent conversion of wind speed to wind power for individual wind turbines and for the super-turbine for quantifying differences.





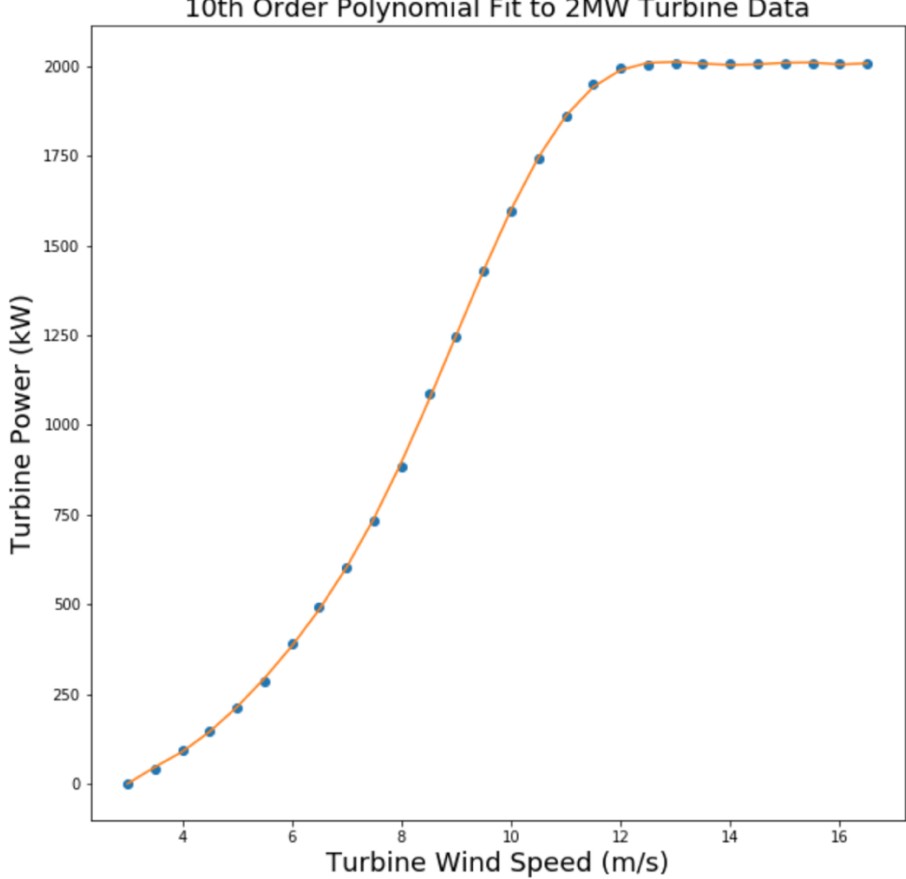

**Figure 2:** 10th-order polynomial fit (orange line) to 2 MW Vestas turbine data from https://en.wind-turbine-models.com/turbines/16-vestas-v90#powercurve.

**2.3 Simulation of a Hypothetical Wind Farm with Low Wind Variability**

The simulation results for the hypothetical wind farm with lower wind variability (standard deviation of 1 m s$^{-1}$) match our hypothesis that at low wind speeds the super-turbine power conversion value is less than the turbine-level power conversion value, while at high wind speeds the super-turbine power conversion value is greater than the turbine-level power conversion value. For each of the 1000 instances we simulated, we used the polynomial equation to convert the wind speed to wind power for each wind turbine as well as the average of the wind speed for the 100 turbines for each instance. Then, we took

the average of the wind power calculated over the 1000 simulated instances for each wind speed mean and standard deviation. The turbine-level mean values are plotted as red asterisks in Figure 3 for mean wind speeds from 6 to 13 m s$^{-1}$ drawn from a Gaussian distribution with wind speed standard deviations of 1 m s$^{-1}$. These are compared to the super-turbine mean averaged across the 1000 instances, which are indicated by blue dots in Figure 3. The plot shows that the mean for the turbine-level power conversion is less than the mean for the super-turbine power conversion for wind speeds at 9 m s$^{-1}$ and

greater. For wind speeds less than 9 m s$^{-1}$, the opposite is true, as is expected from Jensen's Inequality.







## Turbine-Level vs Super Turbine Power Conversion

★ Turbine-Level Mean Power
● Super Turbine Mean Power

**Figure 3:** Comparison of the turbine-level power conversion to the super-turbine power conversion for the hypothetical wind farm with 1 m s$^{-1}$ standard deviation across the turbines.

5    The difference between the turbine-level power conversion and the super-turbine power conversion, shown in Table 1, indicates that the turbine-level power conversion has a greater average value than the super-turbine power conversion up to 8 m s$^{-1}$, but at 9 m s$^{-1}$ and greater the reverse is true with the maximum difference of -61.28 kW per turbine. For a wind farm with 100 turbines with 2 MW capacity each, the super-turbine wind conversion would result in an over-estimate of power by over 6 MW, an error of approximately 3%. For a two-sample related t-test, all wind speeds are significantly different at the
10   95% level as shown in the third column of Table 1.



**Table 1:** Difference of the super-turbine power conversion subtracted from the turbine-level power conversion for the hypothetical wind farm of 100 turbines with 1 m s$^{-1}$ variability simulated 1000 times. The p-value indicates statistical significance at the 95% level for a two-sample related t-test.

| Wind Speed (m s$^{-1}$) | Difference (kW/turbine) | P Value |
|---|---|---|
| 5 | 24.16 | 0.00 |
| 6 | 30.49 | 0.00 |
| 7 | 35.77 | 0.00 |
| 8 | 33.75 | 0.00 |
| 9 | 2.07 | 0.00 |
| 10 | -34.43 | 0.00 |
| 11 | -61.38 | 0.00 |
| 12 | -41.33 | 0.00 |
| 13 | -14.51 | 0.00 |
| 14 | 2.18 | 0.00 |

## 2.4 Simulation of a Hypothetical Wind Farm with High Wind Variability

The simulation results with higher wind variability (standard deviation of 2 m s$^{-1}$) similarly match our hypothesis that at low wind speeds the super-turbine power conversion value is less than the turbine-level power conversion value, while at high wind speeds the super-turbine power conversion value is greater than the turbine-level power conversion value; however, the magnitude of the differences is greater than with lower wind speed variability.

The results for the simulations drawing from a Gaussian distribution with mean wind speeds of 6–13 m s$^{-1}$ and a 2 m s$^{-1}$ wind speed standard deviation appears in Figure 4. For larger variability in the wind across the wind turbines, the deviation is more pronounced between the turbine-level wind power conversion and the super-turbine power conversion, especially at wind speeds of 10, 11, and 12 m s$^{-1}$. The differences between conversion methodologies are shown in Table 2, with the maximum difference of -165.66 kW per turbine at a mean wind speed of 11 m s$^{-1}$. For a wind farm with 100 turbines with 2 MW capacity, the super-turbine wind conversion would result in an over-estimate of wind speed by over 16 MW, an error of more than 8%. For a two-sample related t-test, all wind speeds are significantly different at the 95% level as shown in the third column of Table 2.

**Table 2:** Difference of the super-turbine power conversion subtracted from the turbine-level power conversion for the hypothetical wind farm of 100 turbines with 2 m s$^{-1}$ variability simulated 1000 times. The p-value indicates statistical significance at the 95% level for a two-sample related t-test.

| Wind Speed (m s$^{-1}$) | Difference (kW/turbine) | P Value |
|---|---|---|
| 6 | 30.92 | 0.01 |
| 7 | 83.76 | 0.00 |
| 8 | 82.14 | 0.00 |
| 9 | -6.47 | 0.00 |
| 10 | -120.23 | 0.00 |
| 11 | -165.66 | 0.00 |
| 12 | -137.44 | 0.00 |
| 13 | -40.97 | 0.00 |







**Figure 4:** Comparison of the turbine-level power conversion to the super-turbine power conversion for the hypothetical wind farm with 2 m s$^{-1}$ standard deviation across the turbines.

## 3 Shagaya Wind Farm

5   In addition to the simulated hypothetical data, we examined data from a 10 MW wind farm located at the Shagaya Renewable Energy Park in Kuwait. The location of the wind farm is labeled Shagaya in Figure 5 and the turbines are located at an elevation of 240 meters and approximately a latitude of 29.22°N and a longitude of 47.05°E. This local topography is flat and the climate is characterized by persistent arid conditions with large temperature differences between summer and winter. There are five 2-MW turbines currently located at Shagaya with data available at ten-minute frequency

10   from 1 September 2017 until 31 May 2018. The data include the mean power produced by each turbine over a ten-minute period as well as the standard deviation, minimum, and maximum of the one-minute raw data over this ten-minute period.



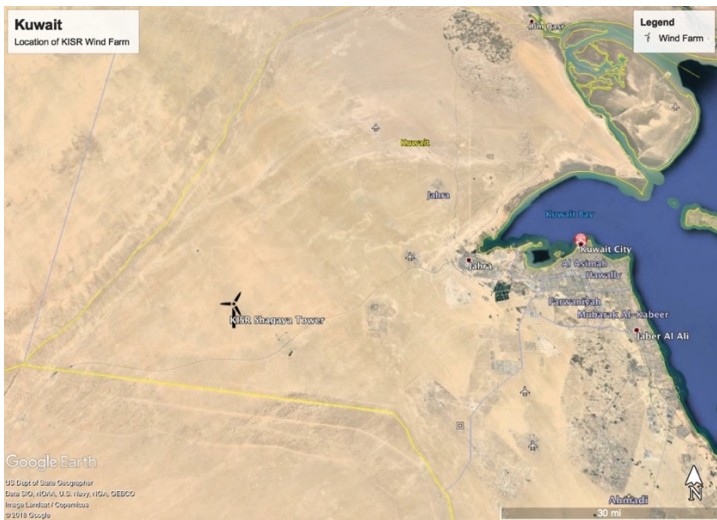

**Figure 5:** Map of the Shagaya wind farm location in western Kuwait characterized by flat, arid desert.

The data were pre-processed for quality control beginning with removing instances with missing data that occurred in approximately 22.5% of the original dataset. Negative power observations in the dataset were set equal to 0 MW power. All values reported above 2020 kW were replaced with 2020 kW since that was approximately the maximum observed. Finally, if the wind speed at the turbine was measured at greater than 3 m s$^{-1}$ but no power was reported, those instances were removed from the dataset as they reflected times of possible maintenance or other forced shutdown of the turbine, which occurred in 2.09% of the original dataset. The total dataset size after quality control had 23,679 instances that included measured power at all five turbines.

Next, we quantified the difference in nacelle wind speed, measured wind power, and converted wind power among the turbines at the Shagaya wind farm. Table 3 shows the mean wind speed (second column) and measured mean power (third column) differences between turbine 1 and all other turbines. The mean difference in wind speed varied from 0.03 to 0.13 m s$^{-1}$ and the mean power differs between 19.27 and 27.19 kW. We then computed the farm-level power with the super-turbine approach by using the mean wind speed across the turbines and using the polynomial fit to convert to power. We also computed the turbine-level total wind farm power by converting the wind speed at each turbine to power and taking the sum across all turbines. Comparing both power conversion techniques to the actual power produced we found a mean absolute difference of 2.63 kW per 2 MW turbine, or a total wind farm power difference of 13.15 kW. We then computed a mean absolute error of 68.83 kW per 2 MW turbine between the super-turbine power conversion and measured power and a mean absolute error of 68.52 kW per 2 MW turbine between the turbine-level power conversion and the measured power.

**Table 3:** Mean wind speed and mean power differences between one turbine and all other turbines at the Shagaya wind farm.

| Difference Between Turbine 1 and… | Mean Wind Speed Difference (m s$^{-1}$) | Mean Power Difference (kW) | Mean Absolute Wind Speed Difference (m s$^{-1}$) | Mean Absolute Power Difference (kW) |
|---|---|---|---|---|
| 2 | 0.13 | 19.27 | 0.28 | 61.02 |
| 3 | 0.09 | 22.38 | 0.31 | 70.16 |
| 4 | 0.03 | 23.24 | 0.35 | 73.80 |
| 5 | 0.08 | 27.19 | 0.41 | 100.11 |

The differences between the power conversion using a polynomial fit to the wind speed data and the measured power are not only due to the effect of Jensen's Inequality, but also due to the Shagaya wind speeds measurements by nacelle anemometers. These measurements occur behind the blades of a turbine and therefore the wind speeds are impacted by

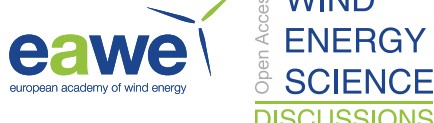



wake effects. St. Martin et al 2017 showed that there is a substantial difference at wind speeds of greater than 9 m s$^{-1}$ and accounted for the wake effects of using nacelle wind speeds for power conversion by applying a fifth order polynomial fit between an upwind met tower and the nacelle wind speed data. We avoid the use of a transfer function to map between a met tower and the nacelle wind speeds because we want to isolate the impact from Jensen's Inequality; however, an operational
power conversion methodology should attempt to take into account the impact from using nacelle wind speeds to convert to power and therefore should include the met tower observations either as a predictor in the power conversion machine learning or should apply a transfer function to the nacelle wind speed data.

Finally, we compared the super-turbine power conversion to the turbine-level power conversion from the mean wind speed
and nacelle wind speeds at the individual turbines. The data are plotted in Figure 6 with the super-turbine mean power per 2-MW turbine on the x-axis and the turbine-level mean power per 2-MW turbine on the y-axis. The scatter along the 1:1 line aligns with the hypothetical data analysis where for wind speeds less than 8 m s$^{-1}$, the super-turbine mean underestimated the power, and for wind speeds greater than 8 m s$^{-1}$ the super-turbine mean overestimated the power.

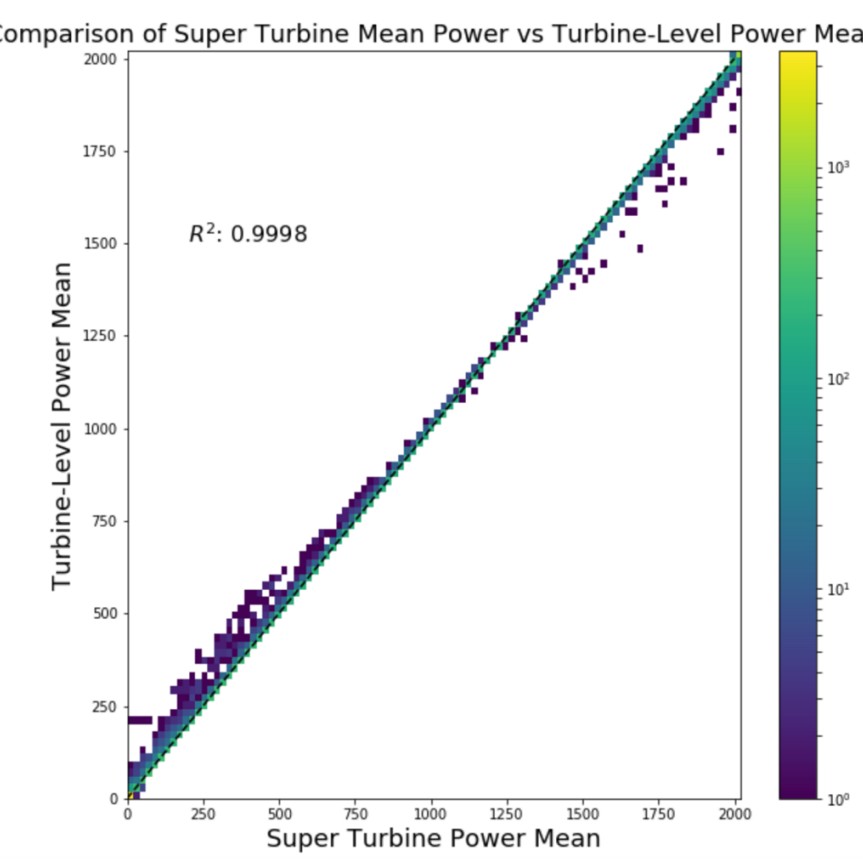

**Figure 6:** Scatterplot of the super-turbine power conversion to the turbine-level power conversion from the mean wind speed and wind speeds at the individual turbines at the Shagaya wind farm.

## 4 Discussion

The greater difference for the power conversion in the 2 m s$^{-1}$ standard deviation hypothetical wind speed variability scenario
of the simulated data is a result of a higher frequency of turbine-level power conversions further from the mean of the wind speed. This is illustrated in Figure 7 where the blue distribution on the x-axis indicates the mean wind speeds used for the super-turbine power conversion, which is shown as the blue distribution on the y-axis. The red distribution on the x-axis

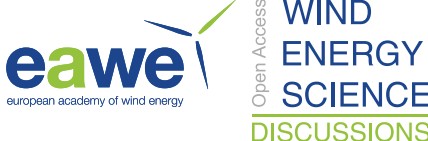

indicates the turbine-level wind speeds used in the conversion to power, which is shown as the red distribution on the y-axis. This analysis is for the 1000 simulated instances in the 2 m s$^{-1}$ variability scenario. The turbine-level power conversion draws from a wider distribution while the mean value in the super-turbine power conversion draws from a narrower distribution. This result occurs because taking the mean of the individual simulations narrows the distribution via the law of
large numbers where the mean of a large number of simulations or observations should approach the expected value as the number of simulations or observations increase (Wilks 2011). The wind speed and wind power are asymmetric around the mean of the distribution of the wind speed at 10 m s$^{-1}$, as illustrated in Figure 1 where the orange line is the polynomial fit to the data for the cubic power transformation. At wind speeds of 12 m s$^{-1}$ and greater, the power stays approximately constant at the maximum value of 2000 kW. However, below wind speeds of 10 m s$^{-1}$ on the left-hand side of the wind speed
distribution, the power decreases according to the orange line and does not hit a minimum value in the same way the power achieves a maximum value on the right-hand side of the wind speed distribution. This simulated dataset illustrates the impact of Jensen's Inequality on the wind speed to power conversion and how larger variability will introduce greater differences due to more samples drawn from the distribution further from the mean.

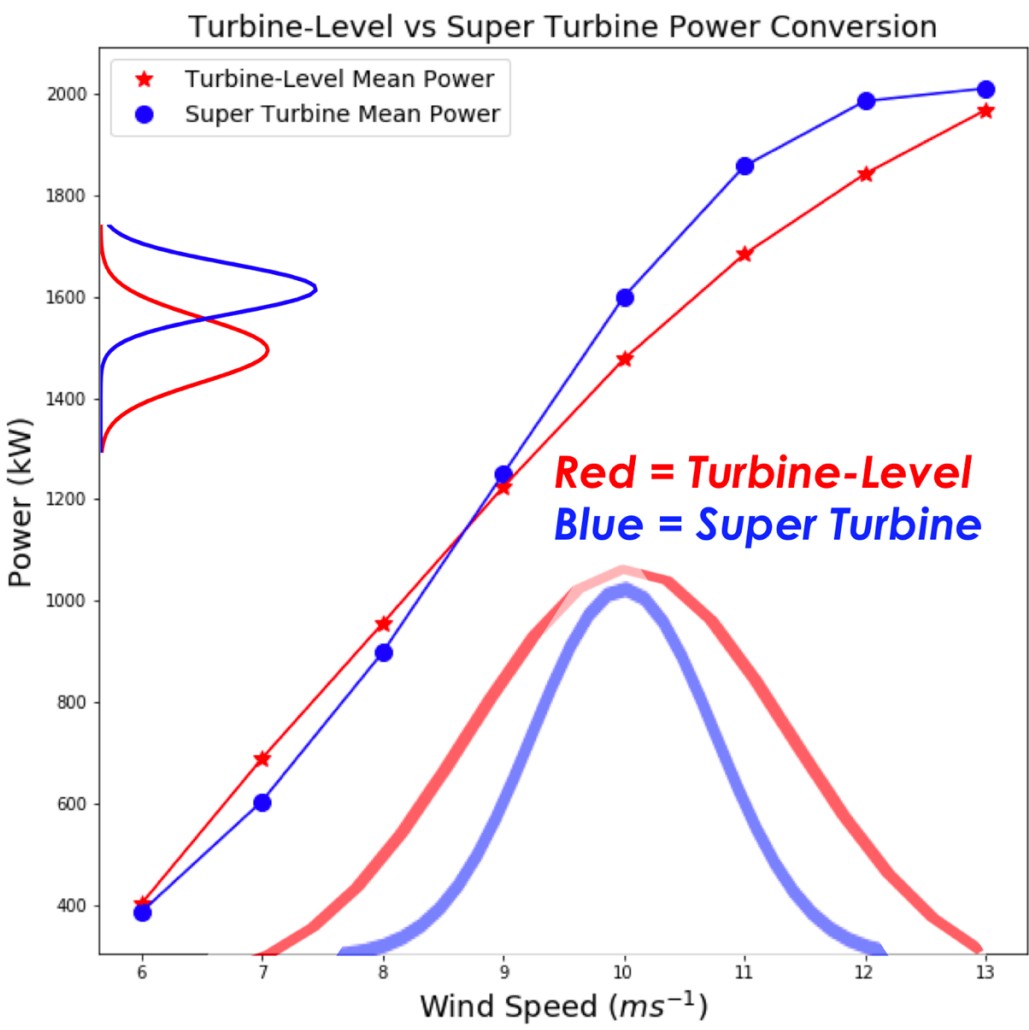

**Figure 7:** Illustration of the distribution of simulated wind speeds for each turbine and the super-turbine mean. The red indicates the super-turbine power conversion distribution for the 1000 simulated instances in the 2 m s$^{-1}$ variability scenario, and the blue indicates the turbine-level power conversion in the same variability scenario.



The empirical data from the Shagaya wind farm in Kuwait highlights the same structural differences between the super-turbine wind power conversion and the turbine-level wind power conversion. Although the magnitude of the differences is less than the magnitude of the simulated hypothetical data, the wind farm in Kuwait is characterized by less variability among the turbines than may be expected from a wind farm that covers a larger spatial area, located in more diverse geography, or that experiences more variable weather that could produce greater wind speed variability among turbines. Wind farms may not measure wind speed and wind power at each turbine individually; therefore, a technique to predict the total wind power at the connection node given information about the mean wind speed and the variability across the wind farm would be valuable, and machine learning may provide an alternative solution.

## 5 Machine Learning

Machine learning has been used to convert wind speed to power for wind farms where data are available (Mahoney et al 2012, Parks et al. 2011). Machine learning is best utilized when there is a non-linear relationship among the predictors and the predictand and the true relationships can be found in the dataset, which is a characteristic of this wind power conversion problem. The machine learning model used here is the random forest supervised learning method (Breiman 2001). The random forest represents an ensemble of regression trees where the final prediction is an average of the prediction from each of the trees. Regression trees utilize the predictive power of dividing a dataset into smaller subsets based on the predictive relationships between the predictor and the predictand until the subsets minimize the cost function (Witten and Frank 2005). Regression trees do not search for the most important predictor in order to split a node, but rather search for the best predictor only among a random subset of the predictors. This technique results in a final model that reduces overfitting the training data and a final model that ultimately generalizes better (Witten and Frank 2005). The random forest used here is the python package *scikit-learn* random forest regressor (Pedregosa et al. 2011).

### 5.1 Hypothetical Wind Farm

Our goal in applying the random forest to the hypothetical data is to show that this machine learning method is able to learn the effect of Jensen's Inequality on the super-turbine wind power conversion from using the mean wind speed rather than the individual wind speeds at each turbine. We use the turbine-level power conversion as the "observed" data that we are trying to predict since this is the power aggregated from each turbine to the total farm level. Random forest models were trained on each simulated wind speed dataset independently, which means that there were 10 random forests for each wind speed from 5 to 14 m s$^{-1}$. The predictors provided to the random forest were the super-turbine power conversion, the mean wind speed and the standard deviation of the wind speed. The optimal random forest configuration was found to have a maximum number of trees of 200. The maximum number of predictors the random forest uses in an individual tree was found to be two, and the minimum number of leaves that are required to split an internal node was determined to be one. We randomly split the dataset into 80% training and 20% testing and all results are shown on the test dataset.

The random forest is able to substantially reduce the error from the super-turbine power conversion for all wind speeds, except for 9 m s$^{-1}$, which is right at the inflection point in the polynomial power conversion and has minimal effect from Jensen's Inequality, as shown by the difference of 2.07 kW per 2 MW turbine prior to applying machine learning. The results are shown in Table 4 where the average difference between the super-turbine power conversion and the turbine-level power conversion appears in the middle column and the mean absolute error (MAE) of the random forest in the right column. For wind speeds less than or equal to 9 m s$^{-1}$, the super-turbine power conversion overestimates the power while the opposite is true for wind speeds greater than or equal 9 m s$^{-1}$. However, the random forest is able to reduce the MAE to between 0.51 and 2.42 kW per 2 MW turbine for all wind speeds. This minimal remaining amount of error could be due to randomness in the simulations of the 100 turbines because we provided the random forest the standard deviation of the wind speeds for the 100 turbines.



**Table 4:** Difference of the super-turbine power conversion subtracted from the turbine-level power conversion for the hypothetical wind farm of 100 turbines with 1 m s$^{-1}$ variability simulated 1000 times in the middle column. The MAE of the random forest for each wind speed is shown in the right-hand column.

| Wind Speed (m s$^{-1}$) | Super-turbine Power Conversion Difference (kW/turbine) | Random Forecast Power Conversion Error (kW/turbine) |
|---|---|---|
| 5 | 24.16 | 2.42 |
| 6 | 30.49 | 0.51 |
| 7 | 35.77 | 0.56 |
| 8 | 33.75 | 1.44 |
| 9 | 2.07 | 2.34 |
| 10 | -34.43 | 2.01 |
| 11 | -61.38 | 1.03 |
| 12 | -41.33 | 1.54 |
| 13 | -14.51 | 1.75 |
| 14 | 2.18 | 1.13 |

## 5.2 Shagaya Wind Farm

We next applied the random forest to the Shagaya empirical data to determine whether we can improve upon the mean difference of 68.83 kW per 2-MW turbine between the super-turbine power conversion and measured power. Once again, we randomly split the dataset into 80% training and 20% testing and all results are shown on the test dataset.

We systematically tested multiple variations of predictors available in order to minimize the error in converting the wind speeds at each farm to the measured power. First, we tested giving the random forest the predictors of the turbine-level converted mean power and the standard deviation across turbines and calculated the MAE of 54.56 kW per 2 MW turbine. Then, we computed the mean wind speed and the standard deviation of the wind speed and used those as predictors along with the super-turbine power, the error reduced slightly to 53.67 kW per 2 MW turbine. Next, we tested using the super-turbine mean power and each turbine's individual wind speed as predictors and found that the MAE was reduced to 50.17 kW per 2 MW turbine. These different predictor sets show that providing the machine learning model each of the individual turbine wind speeds allows the model to better train to the variability across the turbines. Note that since there were only five wind turbines in this dataset, the standard deviation may not adequately represent the variability across the turbines compared to a wind farm with a hundred turbines and likely a more normal distribution with variability better represented by the standard deviation. Finally, we tested adding in the five-turbine mean temporal standard deviation of the one-minute wind speeds over the ten-minute interval as a predictor. This predictor set that included only the individual turbine wind speeds and the temporal standard deviation of the wind speeds produced the lowest error with the MAE decreasing to 42.94 kw per 2 MW turbine. This is a 37.6% reduction in error from the original super-turbine power conversion using the mean wind speed by using machine learning compared to the super-turbine power conversion using the mean wind speed.

Finally, we compared the predictions to the measured power to evaluate the distribution of differences across the range of measured power and found the differences increased as the power increased although the majority of the instances fell along the 1:1 line. The predictions from the random forest are shown on the x-axis with the measured power on the y-axis for Figure 8. The R$^2$ value of 0.9898 for a linear fit to the observed and predicted power highlights the predictive power of the random forest in power conversion. The greater variability in the differences between the measured power and the predicted power at higher values of power could be a function of mechanical reasons that cause the turbines to produce lower power than expected at a given wind speed. Ultimately, however, one would not expect a machine learning model to capture



decreases in power produced when the turbines may not be functioning at rated capacity unless the dataset included information about curtailment or other mechanical causes.

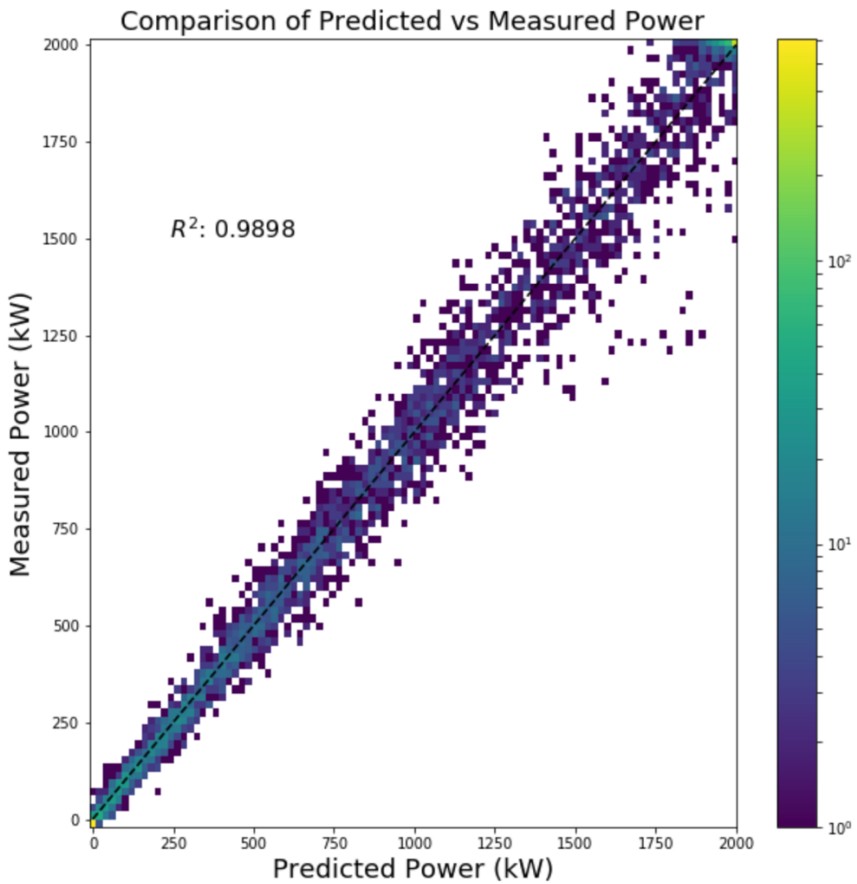

**Figure 8:** Scatter plot of the measured mean power at the Shagaya wind farm compared to the random forest predicted mean, with the black line indicating a linear fit with a $R^2$ of 0.99.

## 6 Conclusions

The wind power forecasted at the farm level is of utmost importance for a utility or system operator; however, the variability at the farm level is a function of the variability across the turbines at the farm. In this study we use both hypothetical
simulated data and empirical data to analyze the effect of Jensen's Inequality on the application of a super-turbine power conversion where a mean wind speed is used to convert to power. We showed that there are systematic non-linear differences between a turbine-level power conversion and a super-turbine power conversion in a range of wind speeds from 5 to 14 m s$^{-1}$. The effect of Jensen's Inequality was found to be most pronounced at approximately 7 and 11 m s$^{-1}$ in the simulated hypothetical wind farm, where the curvature of the power curve is the greatest. Understanding the impact of
Jensen's Inequality on the total power at a wind farm, or the super-turbine wind conversion paradox, allows a utility to choose a power conversion methodology that incorporates this effect for a more accurate power conversion estimate.

In the empirical data analysis, we were similarly able to show differences between the turbine-level power conversion and a super-turbine power conversion even for a relatively small 10 MW wind farm consisting of five individual turbines in flat
desert terrain. One would expect that a wind farm with more turbines in a larger area may exhibit more variability,



especially if there are local terrain or wake effects. In the hypothetical data we showed there is a larger effect of Jensen's Inequality as the wind variability increases as would be expected for a larger wind farm.

Finally, we showed that the random forest machine learning method is able to reduce the error in the wind speed to power conversion when provided predictors that quantify the differences due to Jensen's Inequality. This was first done using the hypothetical simulated data where the error was reduced to under 2.5 kW per 2 MW turbine for all wind speeds from 5 to 14 ms$^{-1}$. In the empirical data analysis, we were able to reduce the error from an average difference of over 68 kW per 2 MW turbine to a little over 42 kW per 2 MW turbine, which is an error reduction of greater than 37%.

In this study, we focused on utilizing machine learning to isolate and remediate the differences caused by Jensen's Inequality on wind speed to power conversion. We did not try to find the lowest error methodology for converting wind speed to power as a machine learning model would likely have lower error when including other meteorological variables such as wind direction, temperature and humidity. The error for the machine learning method would also be impacted by measurement error including the error caused by using nacelle wind speeds without a transfer function, so we would not expect any
method to produce an error of 0 kW per 2 MW turbine in this study. The super turbine approach typically will use power measured at a meter for the entire farm whereas the turbine approach will use the power measure at each turbine; however, there can be a discrepancy between the sum of the turbine powers and the power measured at the farm's meter due to losses in transmission. Ultimately, utilities are interested in the power measured at the farm's meter or at a meter on the transmission line away from the farm and the machine learning method should produce accurate predictions of power at that
meter considering the effect of Jensen's Inequality.

Jensen's Inequality can produce significant differences in wind power conversion between a super-turbine and a turbine-level approach to power conversion, which we have named the super-turbine wind power conversion paradox. This analysis suggests that forecasters responsible for predicting power for a utility should perform power conversion at the turbine level
or use machine learning to reduce the effects of Jensen's Inequality in power conversion. Additionally, if the temporal standard deviation of wind speed is known, machine learning can incorporate both the effects from Jensen's Inequality on the spatial and temporal variability of wind speeds across wind turbines at a wind farm.

*Data and code availability:* The data and code from this study is under NDA and cannot be released.

*Competing interests:* The authors declare no competing interests.


*Acknowledgements:* The authors thank Drake Bartlett of Xcel Energy for bringing attention to this issue. We also wish to acknowledge the Kuwait Institute for Scientific Research (KISR) and Dr. Majed Al-Rasheedi for use of the Shagaya wind farm data. We thank Drs. Jared Lee, Gerry Wiener, and Branko Kosovic for helpful comments. The National Center for Atmospheric Research is sponsored by the National Science Foundation.

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
