# Peer review of "The Super-Turbine Wind Power Conversion Paradox: Using Machine Learning to Reduce Errors Caused by Jensen's Inequality"

_Wind Energy Science, 2018_

## Referee Comment (RC1) · Anonymous Referee #1 · 21 Mar 2019

General comments

The authors in this paper forecasted wind speed to power to provide an estimate of generating capacity in which the effect of the local terrain and array orientation are being investigated. The impact of these parameters is tested with a hypothetical wind farm of 100 2-MW and found that there is about 3% difference due to Jensen's Inequality. The paper interesting and well fit within the scope of 'wind energy science' journal.

[Figure]

Specific comments

The following questions (specific comments) needs to be addressed; 1. The author claims that 'There are two main sources of error in wind power forecasting: the error in the underlying weather forecast (i.e., wind speed, and to a lesser degree air density)'. The impact of air density can't be ignored especially when turbines are in extreme high and low temperature. The air density can also improve the prediction accuracy and thus uncertainty. For more details see following papers, • 'Incorporating air density into a Gaussian process wind turbine power curve model for improving fitting accuracy'. • Wind power curve modeling in simple and complex terrain using statistical models. Needs to include such recent results into the literature review 2. Since this paper is about Jensen's Inequality; a solid literature review on that needed. 3. Why only two std deviations? What result would you get with more std deviation values? 4. How you pre-processed the data of Shagaya wind farms? Any reference or explanations? What kind of data have you used; SCADA? 5. In section 4; Random Forest a machine learning approach is being used. Why you have used this particular technique? Why not say GP, SVM or something else? What motivated you to use this techniques? Does this machine learning approach have any limitation? A reference needs to be included in this method of comparative analysis with other machine learning approach. 6. The author used the python package for random forest model construction but no explanation is given on the descriptions or methodology.

technical corrections

Use of simple sentences, a flow chart to describe the methodology needed. A thorough check on reference styling needs to be checked.

---

## Referee Comment (RC2) · Anonymous Referee #2 · 22 Mar 2019

Review: This paper takes the very interesting paper by Jensen and applies it to the over-prediction or under-prediction of power in wind turbines, with increasing inaccuracy as the Standard Deviation of the wind increases. The paper also looks at this for real turbine data and shows the same effect. The authors then use a machine learning method to remove this effect. I am not sure the machine learning section is necessary but the paper provides a novel contribution and makes for an interesting discussion.

Section 2.3: this analysis would be more interesting with some percentage errors. What is the mean error and what is the maximum error? It might be small for this set,

it looks like it from figure 3, but this could be larger for other datasets. Perhaps some discussion on this would be interesting?

Section 2.4: Can you justify the SD of 2ms$^{-1}$? Is this value high or low for real data sets?

Section 3: It would be good to track how many datapoints there were at the beginning, I can work it out but easier if it is in the paper.

Section 3: I am not sure of setting the powers above 2020kW to a maximum, they should be removed as they are erroneous, or remain as they demonstrate an offset in the reading.

Section 3: I am not sure about setting the negative powers to 0, they should be removed or remain for the same reasons as above.

Section 3: What did the dataset look like? Some statistics to define the scatter and shape of the data would be very beneficial. What is the correlation of the speed to power? While I believe that the change is due to Jensen's inequality it would be nice to ensure that this was the case.

Figure 7 shows some interesting distributions for the data and it would have been interesting to see these earlier.

Section 5: It would be good to better justify the use of random forest regression, what other methods are available and why this one?

Section 5.2: It would be nice to see the training and testing errors to look for over- or underfitting.

Technical corrections: The use of the term super-turbine is not clear, I have not seen this term used before. The definition appears to be hidden in Figure 1, but I am still not sure if it is the authors term or a general term. In addition the exact meaning of the term is not clear. p.1 Line 36: I am not sure this is Artificial Intelligence, more machine

learning. P.2 Line 13: For completeness it would be nice to state that this is only for increasing convex and concave functions. P.3 Line 15: It would be clearer if the line read 6-13 ms$^{-1}$. p.3 Line 22: It is quite clear that Gauss is the correct distribution as this is not a long-term study. The additional comments make the discussion more confusing. p.3 Line 26: A 10th order polynomial fit would seem to be high, it is generally good practice to use the lowest order fit possible, is 10th order really the lowest order that can be used in this scenario? p.9 Line 1: Reference is not formatted correctly. Figure 2: Please use the journal referencing style of the website Figure 6: The quality of this figure seems poor.

---

## Author Comment (AC1) · 3 May 2019

Interactive comments on "The Super-Turbine Wind Power Conversion Paradox: Using Machine Learning to Reduce Errors Caused by Jensen's Inequality"
by Tyler C. McCandless and Sue Ellen Haupt

**Anonymous Referee #1**

General comments
The authors in this paper forecasted wind speed to power to provide an estimate of generating capacity in which the effect of the local terrain and array orientation are being investigated. The impact of these parameters is tested with a hypothetical wind farm of 100 2-MW and found that there is about 3% difference due to Jensen's Inequality. The paper interesting and well fit within the scope of 'wind energy science' journal.

Specific comments
1. The author claims that 'There are two main sources of error in wind power forecasting: the error in the underlying weather forecast (i.e., wind speed, and to a lesser degree air density)'. The impact of air density can't be ignored especially when turbines are in extreme high and low temperature. The air density can also improve the prediction accuracy and thus uncertainty. For more details see following papers, Incorporating air density into a Gaussian process wind turbine power curve model for improving fitting accuracy'. Wind power curve modeling in simple and complex terrain using statistical models. Needs to include such recent results into the literature review

**Response:** We have added in citations (Bulaevskava et al. 2015, Pandit et al. 2018) on the relevance of air density in converting wind speed to power; however, wind speed is the most important variable and our analysis focuses on the impact of Jensen's Inequality on the power conversion.

2. Since this paper is about Jensen's Inequality; a solid literature review on that needed.

**Response:** Multiple additional sentences and references were added to the introduction on Jensen's Inequality.

3. Why only two std deviations? What result would you get with more std deviation values?

**Response:** Our goal was to show how the variability of the wind, quantified as the standard deviation of the wind speed, impacts the wind speed to power conversion and this was accomplished with 1 m s$^{-1}$ and 2 m s$^{-1}$ 1 standard deviations. We did not need to test all wind speed standard deviations to quantify the impact from Jensen's Inequality.

4. How you pre-processed the data of Shagaya wind farms? Any reference or explanations? What kind of data have you used; SCADA?

**Response:** The paragraphs explaining the raw data and the pre-processing has been expanded to provide additional details as suggested by the reviewer.

*Edited Manuscript: In addition to the simulated hypothetical data, we examined data from a 10 MW wind farm located at the Shagaya Renewable Energy Park in Kuwait. The location of the wind farm is labeled Shagaya in Figure 5 and the turbines are located at an elevation of 240 meters and approximately a latitude of 29.22°N and a longitude of 47.05°E. This local topography is flat and the climate is characterized by persistent arid conditions with large temperature differences between summer and winter. There are five 2-MW turbines currently located at Shagaya with data available at ten-minute frequency from 1 September 2017 until 31 May 2018, which was 34,393 instances in the initial dataset. The SCADA dataset includes the mean power produced by each turbine over a ten-minute period as well as the standard deviation, minimum, and maximum of the one-minute raw data over this ten-minute period.*

*The data were pre-processed for quality control beginning with removing instances with missing data that occurred in approximately 22.5% of the original dataset. Negative power observations in the dataset were set equal to 0 MW power since there were small negative values recorded when the wind turbine was not generating power, likely a result of the turbine consuming a small amount of power. Finally, if the wind speed at the turbine was measured at greater than 3 m s$^{-1}$ but no power was reported, those instances were removed from the dataset as they reflected times of possible maintenance or other forced shutdown of the turbine, which occurred in 2.09% of the original dataset. We converted the measured wind speed to a converted wind power using the 10$^{th}$ order polynomial and all wind power values converted from wind speed that were above 2020 kW were replaced with 2020 kW, since that was approximately the maximum observed. The total dataset size after quality control had 23,679 instances that included measured power at all five turbines. Over this period of time, the average power at each turbine ranged between 787 kW and 813 kW and the average wind speed ranged between 7.00 ms$^{-1}$ and 7.13 ms$^{-1}$ with a standard deviation across the turbines of 0.26 ms$^{-1}$. To quantify the correlation between the wind speed and the power measured at each turbine, we computed the $R^2$ between wind speed and power for each turbine independently. Using all data, the wind speed to power $R^2$ was in the range of 0.76 to 0.86 for each of the turbines and when limiting the data to the range of 3-12 ms$^{-1}$, we found the $R^2$ was in the range of 0.79 to 0.90 for each of the turbines.*

5. In section 4; Random Forest a machine learning approach is being used. Why you have used this particular technique? Why not say GP, SVM or something else? What motivated you to use this techniques? Does this machine learning approach have any limitation? A reference needs to be included in this method of comparative analysis with other machine learning approach.

**Response:** The following sentences were added at the end of the Section 5: Machine Learning paragraph introducing the random forest approach. In addition to this explanation we have added an additional Figure and details as noted in the next comment (6.).

*Edited Manuscript: Note that we opted to use the random forest method because it is a machine learning method that captures non-linear relationships between predictors and the predictand, and has the added benefit of avoiding overfitting since it is an ensemble approach. Other machine learning methods such as the artificial neural network or gradient boosted regression trees may work similarly well, however, our goal is not to find the most optimal machine*

*learning approach but rather highlight that machine learning can be used to learn the impact of Jensen's Inequality in this application.*

6. The author used the python package for random forest model construction but no explanation is given on the descriptions or methodology. technical corrections Use of simple sentences, a flow chart to describe the methodology needed.

**Response:** We added Figure 8 to illustrate the structure of the random forest, where the final prediction is an average of the predictions from each tree in the forest where each tree is given a subset of the available predictors and training data

[Figure]

A thorough check on reference styling needs to be checked.

**Response:** The authors have added additional references and reviewed the reference styling.

**Anonymous Referee #2**

Review: This paper takes the very interesting paper by Jensen and applies it to the over-prediction or under-prediction of power in wind turbines, with increasing inaccuracy as the Standard Deviation of the wind increases. The paper also looks at this for real turbine data and shows the same effect. The authors then use a machine learning method to remove this effect. I am not sure the machine learning section is necessary but the paper provides a novel contribution and makes for an interesting discussion.

Section 2.3: this analysis would be more interesting with some percentage errors. What is the mean error and what is the maximum error? It might be small for this set, it looks like it from figure 3, but this could be larger for other datasets. Perhaps some discussion on this would be interesting?

**Response:** The reviewer brings up an interesting point to use the percentage errors in the analysis. However, the end user, typically a utility or systems operator, cares about the error in kW (or MW) of the forecast. The percentage error may give a misleading quantification of the impact the forecast error has, when the actual power generated in kW is directly applicable. We have included some percentages on page 5 to quantify the percentage difference in addition to the power differences, and we have also listed p-values in Table 1.

As an example using rough estimates: if the wind speed is 5m/s, the average power produced is 400 kW, and the mean difference is 25 kW, then the percentage difference would be 6.25%. If the wind speed is 12.5 m s$^{-1}$, the average power produced is 1975 kW, and the mean difference is also 25 kW, then the percentage difference would only be 1.27%. This would be misleading that the forecast is much better at 12.5 m s$^{-1}$ than at 5 m s$^{-1}$ when the end user is evaluating short/long they are in kW. By showing the actual power differences for all wind speeds and pointing out the percentage difference at key points along the distribution, we have provided the adequate information necessary to understand the quantitative impact of Jensen's Inequality.

Section 2.4: Can you justify the SD of 2ms−1? Is this value high or low for real data sets?

**Response:** The dataset that we have is from flat terrain and a limited (5) number of turbines in a consistently dry climate. In this dataset, we calculated a standard deviation of 0.26 m s$^{-1}$ and a mean wind speed of 7.07 m s$^{-1}$. We have added this information in the second paragraph of section 3. Note that this is the standard deviation of all instances, not the standard deviation centered around a specific mean value such as the case in the hypothetical analysis. We would expect substantially higher variability in uneven terrain, for a greater number of turbines, and for more diverse weather patterns. Therefore, we believe that the values of 1 and 2 m s$^{-1}$ are reasonable estimates of the average and high variability scenarios for large wind farms in non-desert climates.

Section 3: It would be good to track how many datapoints there were at the beginning, I can work it out but easier if it is in the paper.

**Response:** Per the reviewer's suggestion, we added that there were 34,393 instances in the initial dataset.

Section 3: I am not sure of setting the powers above 2020kW to a maximum, they should be removed as they are erroneous, or remain as they demonstrate an offset in the reading.

**Response:** This paragraph has been updated to address this comment. The raw power values were not quality controlled to be a maximum of 2020 kW; however, the power values that were converted from wind speed with the polynomial power conversion that were above 2020 kW were replaced by 2020 kW.

*Edited Manuscript: We converted the measured wind speed to a converted wind power using the 10th order polynomial and all wind power values converted from wind speed that were above 2020 kW were replaced with 2020 kW, since that was approximately the maximum observed.*

Section 3: I am not sure about setting the negative powers to 0, they should be removed or remain for the same reasons as above.

**Response:** We added a section to explain why negative values were set to 0 MW. The power is measured such that when the turbine is not generating, there are occasionally small negative values that represent the power the turbine is using when it is not generating. Since we are trying to predict the power being produced by the turbine, it is fair to set negative values equal to zero.

*Edited Manuscript: Negative power observations in the dataset were set equal to 0 MW power since there were small negative values recorded when the wind turbine was not generating power, likely a result of the turbine consuming a small amount of power.*

Section 3: What did the dataset look like? Some statistics to define the scatter and shape of the data would be very beneficial. What is the correlation of the speed to power? While I believe that the change is due to Jensen's inequality it would be nice to ensure that this was the case. Figure 7 shows some interesting distributions for the data and it would have been interesting to see these earlier.

**Response:** We have added information to the second paragraph about the power and wind speeds for the turbines. We also computed multiple variants of the $R^2$ value for the correlation of speed to power. When we used all data, the value ranged between 0.76 and 0.86 for each of the five turbines. When we used only the instances when the wind speed was greater than 3 m s$^{-1}$ and less than 12 m s$^{-1}$ to understand the correlation between the cut-in speed and the maximum power, we found $R^2$ to range from 0.79 to 0.90. All of this information has been added to the paper in Section 3.

*Edited Manuscript: Over this period of time, the average power at each turbine ranged between 787 kW and 813 kW and the average wind speed ranged between 7.00 ms-1 and 7.13 ms-1 with a standard deviation across the turbines of 0.26 ms-1. To quantify the correlation between the wind speed and the power measured at each turbine, we computed the R2 between wind speed and power for each turbine independently. Using all data, the wind speed to power R2 was in the range of 0.76 to 0.86 for each of the turbines and when limiting the data to the range of 3-12 ms-1, we found the R2 was in the range of 0.79 to 0.90 for each of the turbines.*

Section 5: It would be good to better justify the use of random forest regression, what other methods are available and why this one?

**Response:** We added the following sentences at the end of the Section 5: Machine Learning paragraph introducing the random forest approach.

*Edited Manuscript: Note that we opted to use the random forest method because it is a machine learning method that captures non-linear relationships between predictors and the predictand, and has the added benefit of avoiding overfitting since it is an ensemble approach. Other machine learning methods such as the artificial neural network or gradient boosted regression trees may work similarly well, however, our goal is not to find the most optimal machine learning approach but rather highlight that machine learning can be used to learn the impact of Jensen's Inequality in this application.*

In addition, we added a new figure (8) to better explain the structure of the random forest and why this methodology was chosen to avoid over-fitting yet capture the non-linear impact of Jensen's Inequality.

Section 5.2: It would be nice to see the training and testing errors to look for over- or underfitting.

**Response:** We added more details about the configuration of the random forest and re-ran the method to produce the results for both the training and testing dataset and display these errors in section 5.2

Technical corrections:

The use of the term super-turbine is not clear, I have not seen this term used before. The definition appears to be hidden in Figure 1, but I am still not sure if it is the authors term or a general term. In addition the exact meaning of the term is not clear.

**Response:** The term super-turbine is defined in the second paragraph of the introduction as well as in Figure 1. "*Past research has indicated an advantage in using artificial intelligence methods for wind power conversion (Parks et al. 2011), and we further investigate this in the context of the super-turbine approach. In the super-turbine conversion methodology, the wind speed is forecast as a farm-average value, and that wind speed is converted to farm-level power.*"

p.1 Line 36: I am not sure this is Artificial Intelligence, more machine learning.

**Response:** Machine learning is a sub-category of artificial intelligence (see the following for a more detailed explanation: https://en.wikipedia.org/wiki/Machine_learning), and both would technically be correct in this instance. However, we have updated the terminology to reflect the request of the reviewer.

P.2 Line 13: For completeness it would be nice to state that this is only for increasing convex and concave functions.

**Response:** We have added multiple sentences on page 2 to better explain Jensen's Inequality.

*Edited Manuscript: More generally, Jensen's Inequality states that if you have a non-linear function, the average of the function is not equivalent to the function of the average, and the magnitude of this inequality depends on the non-linearity of the function and the variability (Pickett et al. 2015). Jensen published this mathematical proof over 100 years ago, and while there are some fields such as ecology physiology and evolutionary biology that have studied the impact of Jensen's Inequality (Denny 2017), the impact on wind power forecasting and methods that overcome the impact have not been studied. Denny (2017) provides details around the basic concepts of Jensen's Inequality with specific examples in biology.*
*The impact of Jensen's Inequality in wind power forecasting is best illustrated in the steep portion of the curve for converting wind speed to wind power, which is generally taken as a cubic function following the power density function.*

P.3 Line 15: It would be clearer if the line read 6-13 m s$^{-1}$.

**Response:** Since we sampled centered around each integer values, we feel it is more clear to list the specific wind speeds rather than use a hyphen to represent a range.

p.3 Line 22: It is quite clear that Gauss is the correct distribution as this is not a long-term study. The additional comments make the discussion more confusing.

**Response:** We appreciate the suggestion that the Gaussian distribution is the correct distribution, however, other colleagues thought that we needed to explain the scientific reasoning for choosing the Gaussian distribution (since wind speed is generally accepted to have a Weibell distribution); therefore we would like to retain this explanation.

p.3 Line 26: A 10th order polynomial fit would seem to be high, it is generally good practice to use the lowest order fit possible, is 10th order really the lowest order that can be used in this scenario?

**Response:** The choice of polynomial fit depends on the data. When evaluating datasets with significant noise or scatter, it can be beneficial to use a lower order polynomial to avoid overfitting. Since Vestas has provided nameplate capacity at specific wind speed intervals, there is no noise in the data and the $10^{th}$ order polynomial fits these data points well.

p.9 Line 1: Reference is not formatted correctly.

**Response:** The reference formatting has been updated.

Figure 2: Please use the journal referencing style of the website

**Response:** The reference formatting in the Figure title has been updated.

Figure 6: The quality of this figure seems poor.

**Response:** We do not see any difference in quality between this figure and the other figures. Please let us know if the figure quality still appears poor in this updated version.